# Encapsulation of Carvacrol-Loaded Nanoemulsion Obtained Using Phase Inversion Composition Method in Alginate Beads and Polysaccharide-Coated Alginate Beads

**DOI:** 10.3390/foods12091874

**Published:** 2023-05-01

**Authors:** Esther Santamaría, Alicia Maestro, Carmen González

**Affiliations:** Chemical Engineering and Analytical Chemistry Department, Faculty of Chemistry, Universitat de Barcelona, Martí i Franquès, 1, 08028 Barcelona, Spain

**Keywords:** nanoemulsions, low-energy emulsification methods, phase inversion composition, phase diagram, alginate, chitosan, pullulan, encapsulation, kinetics release

## Abstract

Nanoemulsions have been widely studied as lipophilic compound loading systems. A low-energy emulsification method, phase inversion composition (PIC), was used to prepare oil-in-water nanoemulsions in a carvacrol–coconut oil/Tween 80^®^–(linoleic acid–potassium linoleate)/water system. The phase behaviour of several emulsification paths was studied and related to the composition range in which small-sized stable nanoemulsions could be obtained. An experimental design was carried out to determine the best formulation in terms of size and stability. Nanoemulsions with a very small mean droplet diameter (16–20 nm) were obtained and successfully encapsulated to add carvacrol to foods as a natural antimicrobial and antioxidant agent. They were encapsulated into alginate beads by external gelation. In order to improve the carvacrol kinetics release, the beads were coated with two different biopolymers: chitosan and pullulan. All formulations were analysed with scanning electron microscopy to investigate the surface morphology. The release patterns at different pHs were evaluated. Different kinetics release models were fitted in order to study the release mechanisms affecting each formulation. Chitosan-coated beads avoided the initial release burst effect, improving the beads’ structure and producing a Fickian release. At basic pH, the chitosan-coated beads collapsed and the pullulan-coated beads moderately improved the release pattern of the alginate beads. For acid and neutral pHs, the chitosan-coated beads presented more sustained release patterns.

## 1. Introduction

In recent years, it has been reported that essential oils (EOs) have interesting properties for different sectors, such as in medicine or cosmetics, due to the fact of their anti-allergic, anti-inflammatory, or anticarcinogenic properties. It is also worth mentioning their antifungal and antimicrobial properties, as well as their antioxidant properties, making them interesting natural candidates for food preservation [1,2].

Among the best-known oils are carvacrol, thymol, eugenol, linalool, cinnamaldehyde, D-limonene, etc. Each of these is mostly found in a different plant and has different characteristics and properties [2,3]. Carvacrol has been proved as one of the EOs with the strongest antimicrobial and antioxidant properties. Since this compound has a very low solubility in water, a way to add it in aqueous media is in the form of an oil-in-water emulsion (O/W). Nanoemulsions can be defined as emulsions with a droplet size up to 200 nm and low polydispersity, presenting more stability than conventional macroemulsions. These are used to entrap the EOs, finely disperse them in the medium, increase their stability in the environment, and minimise the change in the sensory properties of the products, especially for food preservation and functionalisation [4]. The encapsulation of these nanoemulsions into gel beads immobilises them and offers the controlled release of the EOs.

Nanoemulsions have been widely used to emulsify EOs. As reported by Chaudary et al. [5] in their review, many researchers have obtained food-grade nanoemulsions using high-energy methods, such as sonication, microfluidisation [6,7,8,9], high-pressure homogenisation, and high shear stirring [10,11,12]. Authors who have obtained nanoemulsions using low-energy methods are fewer, although low-energy methods are cost-effective and fast methods for the production of nanoemulsions [13,14,15]. Low-energy methods include processes such as spontaneous emulsification (SE), phase inversion temperature (PIT), and phase inversion composition (PIC). In order to form oil-in-water nanoemulsions (O/W) using the PIC method, the components that form the dispersed phase (i.e., oily components) + surfactants are first mixed, and then the components that form the continuous phase (i.e., aqueous components) are progressively added. For the PIC method, a change in the spontaneous curvature of the surfactant layer during the addition of the continuous phase is required. As a consequence, a phase inversion occurs. Several authors [16,17,18] reported that to obtain small-sized and stable nanoemulsions it is crucial to cross a region of a direct or planar structure—liquid crystal or microemulsion—while adding the continuous phase, with no free oil appearing. Thereby, the oil is already intimately incorporated into the structure. When more water is added to this structure, the spontaneous curvature increases, the structure reorganises into droplets, and the oil is entrapped in them without the requirement of a high amount of energy, resulting in small-sized diameters and a narrow size distribution. For their system, Solè et al. [19] reported smaller diameters using the PIC method compared to using other high-energy consumption methods, such as sonication and high-speed homogenisation. Little information is available on the use of the PIC method to obtain food-grade nanoemulsions for incorporation into edible films or encapsulation to prevent oxidation or microbial activity in fresh foods, such as fruits and vegetables. Chuesiang et al. [20] obtained food-grade nanoemulsions using the PIC method, with a mean droplet size of 100 nm. In a review by Sneha et al. [21], the smallest size reported using this technique was 30 nm. Wakisaka et al. studied O/W nanoemulsion formation using PIC with a natural emulsifier, which showed quite good results in terms of the mean droplet diameter (20–50 nm) [14,15]. As reviewed by Chaudhary et al. [5] and Ozogul [22], no nanoemulsions with droplet size diameters smaller than 20 nm have previously been obtained. The present study contributes to PIC nanoemulsion formation using linoleic acid as a cosurfactant, a primary dietary omega-6 fatty acid. Linoleic acid is not synthesised by the human body and needs to be incorporated into the organism via food. Linoleic acid’s role as a cosurfactant is crucial to form nanoemulsions using the PIC method, because fatty acid increases its spontaneous curvature at the interface when it is neutralised by KOH, forming the linoleate.

Alginate (Alg), a naturally occurring linear anionic hydrocolloid extracted from brown seaweed, has become the most common shell material for encapsulation due to the fact of its biocompatibility, low toxicity, relatively low cost, and mild gelation with the addition of divalent cations [23,24]. Other authors [25,26,27] have provided an overview of the general properties of Alg and its hydrogels, their use in biomedical applications, and perspectives on Alg forming gels in the presence of various divalent cations, such as Ca^2+^, by crosslinking the carboxylate groups of the glucuronate on the polymer backbone. However, there are two major concerns regarding Ca^2+^-Alg beads: (i) their instability in release media due to the fact of Ca^2+^ leaching when it complexes with other molecules and/or due to the fact of salt exchange; (ii) the high porosity of the beads leading to a burst effect or uncontrolled active principle release [28]. As Atencio et al. reported [29], the use of a two-stage hardening procedure for the capsules resulted in improved protection against oil degradation due to the reduction in the porosity of the beads, but release kinetics studies were not performed to evaluate the coating potential in food release systems.

Chitosan (Ch) is a natural polymer extracted from the exoskeleton of crustaceans, as well as from some mushrooms. Chitosan has good biocompatibility and biodegradation properties, and as chitosan-based materials, they are prominent candidates for drug delivery/biomedical applications [30,31,32]. Pullulan (Pul) belongs to the class of microbial-derived biopolymers produced by the black yeast-like fungus *Aureobasidum pullulans*. It contains nine hydroxyl groups on glucopyranose rings of maltotriose [33]. It is nontoxic, biodegradable, biocompatible, edible, tasteless, odourless, noncarcinogenic, nonimmunogenic, and nonmutagenic [33].

The final objective of the present study was the encapsulation of carvacrol-loaded nanoemulsions with ionic complexation using a two-stage hardening procedure, as well as to evaluate the kinetics release. The phase behaviour of the system was studied, determining the regions where nanoemulsions could be prepared using the PIC method. Then, nanoemulsions were encapsulated into alginate beads, which were successfully coated with chitosan and pullulan. The morphology and porous texture of the freeze-dried beads were observed using scanning electron microscopy (SEM). Furthermore, the loading capacity and encapsulation efficiency of the beads were examined. The experimental data were fitted according to the different release models, the influence of the pH on the diffusion exponent (*n*) was analysed, and the release mechanism of carvacrol was further investigated. A link between the release of the active substance of nanoemulsions in encapsulated beads and the used shell material was sought. 

## 2. Materials and Methods

### 2.1. Materials

Food-grade carvacrol (W224502) with a purity ≥98% was purchased from Sigma Aldrich. The synthetic non-ionic surfactant Tween 80^®^ (P1754), potassium hydroxide pellets 85%, food-grade linoleic acid (W800075) (US) with a purity ≥95%, coconut oil (C1758) from *Cocos nucifera*, chitosan (419419) with a >75% deacetylated degree and high M_W_ ≈ 3.1–3.75 × 10^5^ Da, calcium chloride (C1016) that was anhydrous with a granularity ≤7 mm and purity ≥ 93%, Sudan IV (198102), and Brilliant Blue G (27815) were purchased from Sigma Aldrich. Technical-grade sodium alginate (Alg) with a ratio of β-D-mannuronic acid:α-L-guluronic acid = 58.9:41.1, measured using nuclear magnetic resonance (DMX-500, 500 MHz, Bruker, Billerica, MA, USA), and Mn ≈ 668,000 and Mw ≈ 1,750,000, was obtained using size-exclusion chromatography (see below for details on the methods) was purchased from Panreac. Pullulan with a low M_W_ ≈ 4.2 × 10^5^ Da from ITW reagents was used. Water was deionised and further purified by Milli-Q filtration. 

### 2.2. Determination of Equilibrium Phases 

All components were weighed in different tubes, which were sealed, homogenised with a vibromixer, and kept in a thermostatic bath at 25 °C to reach the equilibrium. Liquid crystalline phases were identified using crossed polarizers and small-angle X-ray scattering (SAXS). 

### 2.3. Small-Angle X-ray Scattering

Small-angle X-ray diffraction scattering (SAXS) measurements were used to determine the structure of the liquid crystals obtained during the phase diagram determination. The measurements were performed in a Hecus X-ray Systems GMBH Graz, equipped with a Siemens Kristalloflex 760 (K-760) generator. The temperature of the samples was fixed using a Peltier Anton Paar (25–300 °C) controller. The radiation wavelength was 1.54 nm. 

### 2.4. Nanoemulsion Formation 

For the study of the nanoemulsion formation, 10 g of each sample was prepared. The final water concentration of the emulsions was fixed at 85% *w*/*w*, with the cosurfactant–surfactant mixture (linoleic acid–potassium linoleate)–Tween 80^®^ ((LA-KL)-T80) at 10% *w*/*w* and the oil mixture, carvacrol–coconut oil, at 5% *w*/*w*. First, 1 g of the mixture of surfactants + cosurfactant (Tween 80^®^ and linoleic acid) at the desired composition was weighed in the range 25–45% (LA-KL). Next, 0.5 g of the oil mixture, composed using different carvacrol% in a carrier of coconut oil (25–45%), was added. According to the phase diagram, for each composition different amounts of water with the corresponding potassium hydroxide content was added drop by drop under continuous agitation with a vibromixer until a liquid crystal region or microemulsion region was reached. Several authors [17,18,19,34,35] have reported the importance of reaching a zone with single direct or planar structures without oil excess along the emulsification path for obtaining stable, small-sized O/W nanoemulsions through a low-emulsification method. This mixture was then heated to 70 °C. Different SAXS analyses (data not shown) were performed in order to ensure the presence of a liquid crystal structure at these temperatures. The increase in temperature provided lower viscosities for the crystalline phases so that the incorporation of water and the agitation during the second step were easier. The subsequent second step involved the addition of the remaining water at 25 °C drop by drop into the hot mixture, while the tube was stirred with a vibromixer. The rate of addition is a relevant parameter in the formation of nanoemulsions [17]. Adding the remaining water drop by drop within 2 min led to the formation of the nanoemulsion, while it did not form when the water was added all at once. After the nanoemulsion formed, the mixture was stirred for one more minute. 

### 2.5. Droplet Size

Dynamic light scattering measurements were performed with a nanostructured liquid characterization unit. A 3D dynamic light scattering (3DDLS) spectrometer (LS Instruments, Switzerland) equipped with a He-Ne laser (λ = 632.8 nm) was used, which allowed for temperature control (25 °C), at a scattering angle of 90°. The light intensity correlation function was analysed with the multimodal method, whereas the z-average diameter was obtained using cumulant analysis. The reported droplet size of the nanoemulsions is the mean of at least three independent measurements. 

### 2.6. Stability Measurements 

The stability of the emulsions was determined by studying the variation in the transmittance (T) light (173°) using a Turbiscan Classic MA 2000. To normalise the T values and compare them independently from their initial values, the change in the relative T over time was calculated as: ΔT_rel_ = [(T_t_ − T_t0_)/T_t0_]·100, where T_t0_ is the transmittance at time = 0, and T_t_ is the transmittance at time = t. 

### 2.7. Determination of Molecular Weight (M_W_) of Alg 

The Alg M_W_ was obtained via size-exclusion chromatography using a Waters 2695 separation module with a Waters 2414 refractive index detector and two hydrogel columns of 7.8 × 300 mm at 2000 and 1000 Å (Waters Corp., Milford, MA, USA). Dextran solutions of 2 mg/mL in the range of 80,900–1,800,000 were used for calibration. The dextran and Alg solutions were eluted with 0.1 M NaNO_3_, with a flow rate of 0.6 mL/min at 30 °C. The data were processed using Empower 3 software (Waters Corp.). 

### 2.8. Encapsulation of Nanoemulsions 

The encapsulation of nanoemulsions was carried out using a Buchi Encapsulator B-390. Nanoemulsions with 35% *w*/*w* carvacrol in the oil mixture and 35% *w*/*w* linoleic acid–potassium linoleate in the total surfactant mixture (LA-KL)-Tween 80^®^ were prepared. The nanoemulsions’ final composition was set to 5% *w*/*w* oil mixture, 10% *w*/*w* surfactant mixture, and 85% *w*/*w* alginate aqueous solution at 1% *w*/*w* as a continuous phase. Different beads were produced: alginate-based beads (Alg), gelling of the continuous phase of the nanoemulsion, alginate–chitosan-coated capsules (Alg-Ch), and alginate–pullulan-coated beads (Alg-Pul).

The used encapsulation method was based on that described by Atencio et al. [29], with some modifications. The nanoemulsion was pumped into a nozzle (diameter: 450 μm) with an air pressure of 400 mbar resulting in fluid stream that was sprayed out with a vibration frequency of 40 Hz and an electrostatic field of 350 V to avoid the potential aggregation of the formed beads. The nanoemulsion was dropped into a 1.0% (*w*/*v*) CaCl_2_ solution (corresponding to 0.09 M) for the gelation and formation of the beads and maintained for 10 min with stirring at 200 rpm for hardening. The dropping flow rate was ~0.9 mL/s, which caused a necklace-shaped uninterrupted flow, as recommended for the Encapsulator. All alginate beads were left in the CaCl_2_ bath for 10 min to ensure proper gelation. The beads were collected, filtered, and washed. For the Alg-Ch-coated beads, the beads were hardened using the previously described method of Chew et al. [36]. The method involved a two-stage procedure in which the beads were incubated into a 1.0% (*w*/*v*) CaCl_2_ solution for 10 min and, subsequently, washed and filtered, as described above, followed by 10 min of incubation in 0.1% *w*/*w* Ch solution. For the Alg-Pul-coated beads, once filtered and washed, they were incubated in 6% *w*/*w* Pul solution for 10 min.

### 2.9. Encapsulation Efficiency

To evaluate the encapsulation efficiency, all beads were removed from the hardening bath, and they were washed with the same weight of water, i.e., 50 g of beads was filtered and washed with 50 g of water. The CaCl_2_-filtered solution and washing water were mixed, and the amount of carvacrol was measured using UV–VIS spectrophotometry. 

The encapsulation efficiency (EE) was calculated: (1)$$EE\left(\%\right)=\frac{M_{co}-Mcw}{M_{co}}\times 100$$
where M_co_ is the total carvacrol mass used in the nanoemulsion, and M_cw_ is the remaining mass of carvacrol in the CaCl_2_ solution plus the mass of the carvacrol in the washing water after the washing step. 

The loading capacity:(2)$$Loading\;capacity\left(\%\right)=\frac{M\;car\;in\;microspheres\;(g)}{Microsphere\;mass\;sample\;(g)}\times 100$$

### 2.10. Release Kinetics of Carvacrol Nanoemulsion-Loaded Alg- and Polysaccharide-Coated Alg-Beads 

The release kinetics of the Alg, Alg-Ch, and Alg-Pul beads were analysed. A total of 0.5 g of the beads was put in 200 mL of buffer solution (pH: 2, 7, and 12). The system was then stabilised at 25 °C with mild magnetic stirring (50 rpm), and 2 mL of supernatant was extracted regularly. Subsequently, 2 mL of buffer solution with the same pH was added to the corresponding system. The carvacrol concentration was detected at 276 nm [37] using a UV spectrophotometer. 

The release kinetics of carvacrol from the Alg, Alg-Ch, and Alg-Pul beads were analysed using the Higuchi model (3), first-order model (4), Korsmeyer–Peppas model (5), Baker–Lonsdale model (6), and Gallagher–Corrigan model (7) [38,39]. 

For all models, M_t_ is the quantity of carvacrol released at time t, and $M_{\infty }$ is the quantity of carvacrol released at the end of the experiment (24 h), so $\frac{M_{t}}{M_{\infty }}$ is the fraction of carvacrol released.
(3)$$\frac{M_{t}}{M_{\infty }}={k_{H}t}^{1/2}$$
where k_H_ is Higuchi release constant.
(4)$$ln\left(1-\frac{M_{t}}{M_{\infty }}\right)=k_{1}t$$
where k_1_ is the first-order release constant.
(5)$$\frac{M_{t}}{M_{\infty }}={k_{p}t}^{n}$$
where k_p_ is a constant, and *n* is the release exponent. For 0 < n < 0.45, the release regime is hindered by Fickian diffusion; for n = 0.45, it is Fickian diffusion, and for 0.45 < n < 1, an anomalous transport occurs.
(6)$$\frac{3}{2}\left[1-{\left(1-\frac{M_{t}}{M_{\infty }}\right)}^{2/3}\right]\frac{M_{t}}{M_{\infty }}=k_{BL}t$$
where k_BL_ is the release rate constant. This model assumes the sphericity of the matrix.
(7)$$M_{t}=M_{\infty }\left(1-e^{{-k}_{1}t}\right)+(M_{\infty }-M_{b})\left(\frac{e^{k_{2}t-k_{2}t_{2max}}}{1+e^{k_{2}t-k_{2}t_{2max}}}\right)$$
where M_b_ is the carvacrol released during the first stage (i.e., burst effect), k_1_ the first-order kinetic constant, k_2_ the kinetic constant for the second-stage matrix degradation, and t_2 max_ is the time for the maximum release rate.

## 3. Results and Discussion

In order to study the carvacrol–coconut oil/Tween 80^®^–(linoleic acid–potassium linoleate)/water system for forming O/W nanoemulsions, composition variables should be set. The % *w*/*w* cosurfactant in the total surfactant content, linoleic–linoleate/(Tween 80^®^ + linoleic–linoleate), % *w*/*w* carvacrol in the oil mixture, and neutralisation degree of linoleic acid have an influence on the nanoemulsions’ properties, so their effects were studied. In order to set the neutralisation degree used in this work, preliminary experiments were carried out. As can be seen in Figure 1, several tubes were prepared with different neutralisation values and different carvacrol % *w*/*w*, setting (LA-KL) at 30% *w*/*w*, where (LA-KL) is the % *w*/*w* linoleic acid–potassium linoleate in the total surfactant mixture. 

The nanoemulsions with a small droplet size could be roughly identified since they were translucent or bluish. Therefore, the proper range of the % *w*/*w* carvacrol to be used could be determined by eye, as when a white aspect was observed, the emulsions were coarse. It can be seen in Figure 1 that as the neutralisation value increased, the % *w*/*w* carvacrol range that allowed for the formation of small-sized nanoemulsions, which presented a transparent–translucent appearance, increased in value. The range for the formation of nanoemulsions was approximately 15–40% carvacrol in the oil mixture for a 75% neutralisation degree for the stoichiometric one. When a 100% neutralisation degree was used, the range for the nanoemulsion increased to 20–50% carvacrol. Finally, for a 125% neutralisation degree, the range for the nanoemulsion was 40–50% carvacrol. Subsequently, the stability of the nanoemulsions when measured as being in the middle of each of the three ranges was measured. It was observed that the experiments carried out at a 125% degree of neutralisation of linoleic acid had very poor stability. In contrast, the experiments performed at 75% and 100% degrees of linoleic acid neutralisation had higher stability, and they were very similar to each other. 

To select the neutralisation percentage for the rest of the experiments, the two factors mentioned above, stability and percentage of carvacrol that allows for the formation of the nanoemulsion, were considered. Firstly, the option of working with the value of 125% was discarded due to the low stability and the narrow range of % carvacrol for the formation of small-sized nanoemulsion droplets (only 40–50%). Secondly, between the two remaining options, it was decided to choose the value of 100% due to the fact that despite having very similar stabilities, the % *w*/*w* carvacrol needed to form the nanoemulsion was higher. The main objective of forming nanoemulsions containing carvacrol was to take advantage of its antioxidant and antimicrobial activity. So, the more carvacrol loaded into the nanoemulsions, the better the antioxidant and antimicrobial properties [37,40,41]. 

### 3.1. Nanoemulsion Formulation

In order to study the formation of carvacrol-loaded small-sized nanoemulsions, a factorial experimental design of 3^k^ was performed. Two variables were studied at three levels, carvacrol % and (LA-KL) %. For this study, the range of the carvacrol % in the oil mixture was between 25 and 45%, and the (LA-KL) % *w*/*w* of the surfactant mixture was between 25 and 45%, because outside of these selected ranges coarse, white emulsions with poor stability were obtained. The different emulsification paths started with a fixed amount of water of 10% *w*/*w*, and the water was added progressively to a final composition of 85% *w*/*w* water, with the relationship between the oil mixture and cosurfactant + surfactant mixture being constant during the nanoemulsification process. Thereby, the pseudo-components of the oil mixture, LA-KL, Tween 80^®^, and water occupied the tetrahedron vertexes in the phase diagram, as shown in detail in Figure 2 (right). In all of the three diagrams (Figure 2, Figure 3 and Figure 4), the oil mixture/(cosurfactant + surfactant) mass relationship was 0.5/1. So, the triangular diagram corresponded to a tetrahedron cut, as shown in Figure 2 (right), with the inferior side of the triangle parallel to the edge of the Tween 80^®^-(LA-KL) of the tetrahedron and the superior vertex of the triangle coinciding with the tetrahedron vertex. The relationship between the carvacrol and coconut oil showed an influence on the nanoemulsions’ mean droplet diameter and stability. Without carvacrol, the nanoemulsions could not form and neither could they with more than 45% *w*/*w* carvacrol. Hence, as the carvacrol % was studied as a variable, three phase diagrams with three different carvacrol % were performed with carvacrol values of 25%, 35%, and 45%. Five emulsification paths using 15%, 25%, 35%, 45%, and 55% (LA-KL) % *w*/*w* were studied in each diagram. The dotted lines in the diagrams were used to ease the visualisation and comprehension of the different phasic zones. Figure 2 shows the phase diagram for carvacrol 25% in the oil mixture and an LA neutralisation degree of 100%. The paths studied there correspond to 15%, 25%, 35%, 45%, and 55% (LA-KL). The red lines frame the three emulsification paths ((LA-KL) 25%, 35%, and 45%) used in experiments 1, 2, and 3 of the experimental design (Table 1).

In Figure 2, an inverse microemulsion zone can be observed in the regions with the lowest amount of water, with water inside the swollen reverse micelles, which gradually turned into a bicontinuous microemulsion for a water content of ~30% *w*/*w* and into a direct microemulsion for a higher % water. The transition between reverse and bicontinuous and direct does not imply phase separation, so a single microemulsion zone was set. As the (LA-KL) % *w*/*w* increased, different liquid crystal structures were observed, with a progressively lower interfacial curvature. 

To understand the behaviour of this system, it is necessary to analyse the evolution of the hydrophilic–lipophilic balance (HLB) of the surfactant mixture. The higher the HLB, the higher the hydrophilicity and surfactant layer spontaneous curvature of the formed structures [42]. Tween 80^®^ had an HLB = 15, linoleic acid had an HLB = 1, and potassium linoleate, formed by the neutralisation reaction of linoleic acid with potassium hydroxide, had an HLB = 20. Since linoleic acid is a weak acid, not all of the linoleic acid reacted with the available potassium hydroxide. To determine the degree of conversion, the pH was measured; in this way, it was possible to know the linoleic–linoleate relationship and, therefore, the HLB of the surfactant mixture was calculated as HLB_surf_ = ∑x_i_·HLB_i_, where x_i_ and HLB_i_ are the mass fraction and HLB of each surfactant in the surfactant mixture. In this case, working with 100% neutralisation, the HLB value for the (LA-KL) was low, approximately 7.4, compared to the high value for Tween 80^®^. Therefore, the lower the (LA-KL)%, the higher the HLB value for the surfactant–cosurfactant mixture. The HLB values for the (LA-KL)-Tween 80^®^ mixture were 13.9/13.1/12.3/11.6/10.8. The higher the HLB, the higher the spontaneous curvature of the structures, so planar structures such as lamellar liquid crystal become structures with a more positive curvature, such as hexagonal or direct discontinuous cubic liquid crystals. As the amount of water increased, the effect of dilution and the hydration of the hydrophilic head of the surfactants also increase the curvature. At approximately 70% water, the micelles retained in the cubic liquid crystal separated and could freely move, it was possible to obtain direct microemulsions, and finally, a zone with excess oil appeared. In this zone, the nanoemulsions were prepared. As Solè et al. [34] reported, nanoemulsions can be obtained by diluting other equilibrium phases where all of the oil mixture is incorporated into the structures, such as cubic liquid crystal or direct microemulsions, which is when the dilution occurs; then, all of the oil is already intimately mixed inside the structure and can be easily entrapped in nanoemulsion droplets. However, when an excess of oil appears before the change in the curvature promoted by the addition of water, it cannot be properly emulsified with a low-emulsification method.

For (LA-KL) 15% *w*/*w* and 55% *w*/*w*, no stable nanoemulsions could be obtained. For both emulsification paths, at relatively low concentrations of water (60% *w*/*w*), an excess of oil appeared. This behaviour can be observed in Figure 3 and Figure 4. Although for (LA-KL) 25% *w*/*w*, nanoemulsions were able form, and it can be observed in Figure 2 that at 60% water, free oil is present, and this led to a higher mean droplet diameter than the other formulations and relative instability (Table 1).

Figure 3 and Figure 4 show phase diagrams for carvacrol 35% *w*/*w* and 45% *w*/*w* in the oil mixture. As shown in Figure 2, along the emulsification paths and inside the red lines regions appeared where the oil was completely incorporated into the liquid crystal and/or microemulsion, and there was no observed excess oil. Using these emulsification paths, nanoemulsions could be prepared. In Figure 3, they correspond to experiments 4–8 of the experimental design, and in Figure 4, they correspond to experiments 9–11 (Table 1). 

An experimental design was used in order to obtain information on which composition variable had a significant influence on obtaining stable, small diameter nanoemulsions, as well as to obtain an optimal formulation in terms of the droplet size and stability. As the phase diagrams show, multiple variables affected the resulting nanoemulsions. The experimental design provided an easy way to determine the influence of each variable and their possible interactions, as the optimum of each variable moved depending on the level of the other variable. A factorial 3^k^ + central point design was used, with the number 3 indicating that three levels were studied for each variable, and k = 2 is the number of input variables studied, i.e., the carvacrol % (from 25% to 45%) in the oil mixture and (LA-KL) % (from 25% to 45%) in the surfactant mixture. The central point was replicated three times.

Table 1 shows the 11 experiments developed for the factorial experiment design, where the input composition variables are shown, as well as the polydispersity index (PDI), output variables to be optimised, mean droplet size, and the change in relative transmittance over a period of seven days for the obtained nanoemulsions. For experiments 1–3, corresponding to the phase diagram in Figure 2, the mean droplet size was smaller for (LA-KL) % *w*/*w* 35% and 45%. These two emulsification paths presented single monophasic regions that extended to more than 60% water. The same behaviour was observed for the nanoemulsions that formed using the emulsification paths shown in Figure 3 (experiments 4–8), where small diameters (13–21 nm) were obtained when the emulsification paths crossed regions without an excess of oil at a high %water (70%). This behaviour is confirmed in Figure 4. Experiment 10, with 35% (LA-KL), provided a small-sized nanoemulsion formation (21 nm). In this case, a single direct microemulsion region was found for an approximate water content of 70% *w*/*w*, while for the other contents of (LA-KL) tested (15%, 25%, 45%, and 55%), excess oil existed at 70% water. For experiment 11, corresponding to an emulsification path of (LA-KL) 45% *w*/*w*, no single-phase region appeared, and the droplet size obtained was much larger. 

Morales et al. [35] used the phase inversion temperature method (PIT) to prepare nanoemulsions, where a change in the spontaneous curvature of the structures is promoted by a change in the temperature. They reported that the main requirement for the formation of small-sized O/W nanoemulsions is to achieve the complete solubilisation of the oil in a single phase along the emulsification path. Then, when the change in the curvature takes place to form the nanoemulsion—due to the fact of dilution (PIC) or a change in the temperature (PIT)—the oil is already intimately entrapped in the surfactant aggregates and does not need to be dispersed by a high shear. Therefore, low-energy emulsification methods can be used successfully. If an excess of oil appears, it cannot be properly emulsified by gentle agitation, since it cannot be incorporated into the droplets. Other authors who also used the PIC method came the same conclusion [19,34,35]. Our results agree with those previously reported, as nanoemulsions cannot form if a region without excess oil is not crossed along the emulsification path. However, in the system studied here, another requirement seemed to influence the droplet size of the formed nanoemulsions. Smaller-sized nanoemulsions were obtained when the phase without excess oil was present at higher percentages of water. This is probably related to the fact that, in these cases, less excess oil exists in the final formulation of the emulsions. At 85% water, excess oil was always present. However, the amount of excess oil was lower when there was a zone with all of the oil dissolved near the final composition than when it was far; therefore, it was easier to emulsify. 

As shown in Table 1, very small droplet sizes, smaller than 20 nm, were obtained in some of the experiments, such as those corresponding to the experimental design’s central point (carvacrol 35% *w*/*w* and (LA-KL) 35%), with values of 17, 16, and 18 nm for each of the repetitions for the sample. Very small values were also obtained for the sample corresponding to carvacrol 45% *w*/*w* and (LA-KL) 35% *w*/*w*, with a value of 13 nm.

It should be noted that the nanoemulsions were made with 5% *w*/*w* oil mixture, and sizes smaller than 20 nm were obtained. As Chaudhary et al. [5] reviewed, such small droplet sizes have not been obtained before with such a significant amount of oil mixture through PIC. In fact, few high-energy emulsification methods can obtain nanoemulsions with very small droplet diameters. Yesim Ozogul et al. [22] reported on several nanoemulsion systems, and only one of them had droplet sizes smaller than 20 nm with 5% *w*/*w* oil, in this case through ultrasonication, but no other case has been found using low-energy methods [22]. In many cases, small sizes have been obtained using high-energy emulsification methods but not nearly as small as 20 nm; for example, Francesco Donsi [43] obtained sizes close to 50 nm in a 4% oil phase system, using high-pressure homogenisation, with carvacrol and peanut oil as the dispersed phase. 

### 3.2. Optimisation of Composition Variables 

Once the phase diagrams were studied and the composition range in which nanoemulsions could be obtained was delimited, an experimental design was carried out in order to determine the optimal formulation that allowed to obtain the minimum droplet size and the maximum stability. Phase diagrams showed that multiple composition factors affected the size and stability of the nanoemulsions. A three-level factorial (3^k^ ) + central point design was selected, where k = 2, with the (LA-KL) % in the surfactant mixture and the carvacrol % in the oil mixture as the studied variables. The total compositions of oil, surfactants, and water were kept the same as those used in the previous section, i.e., 5% oil, 10% total surfactant, and 85% water. Table 1 shows the developed experiments, and Figure 5 shows the response surfaces for the mean diameter and the stability, calculated as relative change in the transmittance, of the obtained nanoemulsions. 

The response surface for the nanoemulsions’ diameters is shown in Figure 5a. A minimum diameter can be observed. This minimum diameter shifts towards higher carvacrol % values as (LA-KL) decreases. The optimal (LA-KL) also decreased when the carvacrol % increased. The influence of one variable depended on the value of the other. The minimum diameter was found at values close to 35% (LA-KL) for the cases of 25% and 35% carvacrol, which is in agreement with that already discussed in Section 3.1. In the phase diagram shown in Figure 2, it can be seen that in the case of 25% carvacrol the emulsification path for 35% (LA-KL) crossed long direct zones without free oil (i.e., hexagonal and cubic liquid crystal) that extended to more than 60% water. For the case of 35% carvacrol in the oil mixture (Figure 3), the emulsification path of 35% (LA-KL) passed through a large zone where phases appeared without free oil (i.e., a lamellar liquid crystal region followed by a mixture of lamellar liquid crystal + direct microemulsion, and a direct microemulsion region) that extended to more than 70% water. In contrast, for the other values of (LA-KL), excess oil appeared in equilibrium with lamellar liquid crystal (45% (LA-KL)), or the single-phase zone did not extend to a high % water (25% (LA-KL)). According to Figure 4, for the emulsification path of 35% (LA-KL), all of the oil remained dissolved with a high amount of water in contrast to the other paths tested for 45% carvacrol. It also corresponds to the smaller droplet size obtained for 45% carvacrol.

For the stability response surfaces, as shown in Figure 5b, a global minimum for (LA-KL) 35% *w*/*w* can be observed. The stability of a nanoemulsions depends on the mean size and polydispersity, as a small droplet size slows down the sedimentation, and a small polydispersity slows down the Ostwald ripening, but it is also directly related to the agreement of the HLB of the surfactant mixture with the preferred HLB for the oil mixture. Aulton [44] concluded that stable O/W emulsions are best formulated with emulsifiers or a combination of emulsifiers with HLB values close to that required for the oily dispersed phase. The required or preferred HLB only depends on the composition of the oil mixture. The preferred HLB of the oil mixture can be calculated as preferred HLB_mixture_ = ∑x_i_·preferred HLB_i_, where x_i_ and preferred HLB_i_ are the mass fraction and preferred HLB of each oil in the oil mixture, which is tabulated. Therefore, stable emulsions are best formulated with emulsifiers or combination of emulsifiers having HLB values close to that preferred by the oil mixture. In this case, the preferred HLBs for the oil mixture were 12, 12.4, and 12.8 for 25%, 35%, and 45% carvacrol in the oil mixture, respectively. On the other hand, the HLBs for the surfactant mixture were 13.9, 13.1, 12.3, 11.6, and 10.8 for 15%, 25%, 35%, 45%, and 55% of (LA-KL) in the surfactant mixture. It is possible to observe how the values with the highest similarity are 12.3 of 35% (LA-KL) in the surfactant mixture with 12.4 of 35% carvacrol in the oil mixture. These values correspond to the nanoemulsion that presented the greatest stability, with the smallest change in ΔT_rel_ (%) (central point in the experimental design, experiments 4–7 in Table 1). Presumably the carvacrol molecule, which has a phenol group, is preferentially located on the surface of the oil droplet, with the phenol group oriented to the water, as hypothesised by Gong et al. [45]. Carvacrol would provide some flexibility to the interphase, separating the charged or the highly hydrated headgroups of the surfactants with a high HLB (linoleate and Tween 80^®^), and allowing for a stronger compaction of the surfactant mixture. 

### 3.3. Encapsulation of the Nanoemulsions in Alg, Alg-Ch, and Alg-Pul Beads

As described in Section 2.8, the nanoemulsions were encapsulated in Alg beads coated with different materials. The beads were formed using the same Encapsulator (Buchi B-390) and the same operating variables used by Atencio et al. [29]. The mean diameter was ~1600 μm (*p* < 0.05), measured microscopically. No significant differences between the mean values were observed for the uncoated and coated beads (*p* < 0.05). It is well reported [29,37,46,47] that single Alg gel beads have high permeability and porosity and poor mechanical properties that diminishes their protection and controlled release abilities. In order to improve Alg hydrogels, Wang et al. [46] proposed the inclusion of carboxymethyl–chitosan–copper oxide nanoparticles in the Alg matrix. Cheng et al. [37] reported the use of a mixture of carboxymethyl chitosan and sodium alginate, crosslinked by citric acid with subsequent cooling to −80 °C. Both studies obtained an improvement in the protection properties and a slower release that were pH dependent. In fact, the release accelerated at a basic pH due to the fact of the swelling and disruption of the matrix. In the present work, the Alg beads were reinforced using a simpler method based on that in [29]. Once the alginate beads formed, some of them were set apart, and the others incubated in Ch and Pul solutions, as described above. The encapsulation efficiency was 78%, and the total carvacrol loaded into the beads was 3% (g of carvacrol/g beads). In Figure 6, SEM micrographs show the morphology of the different kinds of beads after being freeze-dried. Before freeze-drying, all of them presented a diameter of 2 mm and were spherical, but after the process it can be seen how the Alg beads shrank (Figure 6a). The Alg beads lost their spherical shape, while, as can be seen in Figure 6d,g, the Alg-Ch and Alg-Pul beads kept their spherical form. The Ch and Pul coating process provided a protective envelope preventing the loss of the shape of the beads, due to the high permeability of alginate, and promoting a more homogeneous surface. High-magnification micrographs ((b) and (c)) show that the Alg beads presented a rough surface, which seems to indicate a high porosity, as other authors have also observed [28,29], while the Ch- and Pul-coated beads showed smoother surfaces than the Alg beads (Figure 6e,f,h,i). The surfaces of the Alg-Ch and Alg-Pul beads were denser than the Alg beads. Moreover, no obvious pores could be seen, especially with the Ch-coated beads. Therefore, the coated beads seem to be better candidates to provide good protection on the loaded nanoemulsion. 

The Ch- and Pul-coated beads presented smoother surfaces, indicating the formation of a layer that compensated for the porosity of Alg. From the micrographs in Figure 6, the Alg-Ch beads (Figure 6e,f) seemed to be more compact than the Alg-Pul beads (Figure 6h,i), because the interaction between Ch and Alg was stronger than the Alg-Pul interaction. The layer was better formed when Ch was used, probably due to the fact that Ch is anchored by COO−NH_3_^+^ bonds to the Alg molecules [48,49], forming a more cohesive and uniform layer than Alg-Pul, which cannot have strong interactions. 

### 3.4. Carvacrol Kinetics Release

To determine the suitability of the beads for the delayed release of carvacrol under different pH conditions, the in vitro release behaviour in buffer solutions at 25 °C was studied. Figure 7 shows the cumulative carvacrol release patterns of the Alg, Alg-Ch, and Alg-Pul beads under different pH conditions. The drug release study was carried out in duplicate for all experiments at pH = 7, one experiment at pH = 2, and one at pH = 12 in order to evaluate the reproducibility of the patterns. The cumulative release results of the duplicates were similar, indicating that results are reproducible.

Nazli and Açikel [48] reported that the resveratrol released from alginate beads was positively correlated with the pH, with the release rate being higher as the pH increased. This is attributed to the fact that, under acidic conditions, the COO− of alginate transformed into –COOH, and the ionisation degree was reduced [49], which led to the contraction of the alginate molecular chain so that the shell of the hydrogels was denser and the release was more efficiently prolonged, whereas the degree of the dissociation of the Alg–COOH groups increased with an increasing pH, which led to stronger repulsion among the single –COO− groups. This behaviour promoted the swelling of alginate beads and an increase in the porosity. This pattern is confirmed in other studies [37,50,51]. In agreement with their results, the release from Alg beads was faster when the pH increased. Accordingly, the beads were swollen at the end of the experiments that had a basic pH.

In Figure 7, it can be observed that the Alg beads presented a rapid release with a strong burst effect. As explained above, in order to minimise this, the alginate beads were coated with chitosan and pullulan to study the effect of each biopolymer. As can be observed in Figure 7 for the experiments with pH = 2 and pH = 7, the Alg beads presented more released carvacrol during the first 200 min. For pH = 2, when pullulan and chitosan were used to coat the Alg beads, the release rate decreased, producing a slower, constant, and more controllable release over time. Therefore, the burst effect was minimised by the layer of another polymer around the Alg beads, protecting the carvacrol for longer. As less carvacrol was released during the first 24 h, the functional properties of the beads conferred by carvacrol were presumably better maintained. The carvacrol retained in the beads could act as a reservoir to provide sustained antimicrobial and antioxidant properties to the medium—for example, a food—in which the beads were added. For pH = 7, the Alg-Ch beads presented a slower release rate, and after 24 h the amount released was 47%, followed by 64% for the Alg beads and 59% for the Alg-Pul beads. For pH = 12, the release patterns were inverted, and the Alg-Ch beads not only presented the highest release rate, but the beads totally disintegrated under basic conditions. Some authors related this behaviour to the formation of nanocolloidal complexes of Alg-Ch. Wassupalli et al. reported that the complex formation was stable at a pH range of 3.5 to 8.5 [52]. At a pH above 8.5, the chitosan collapsed and broke down the Alg-Ch beads. 

The Alg and Alg-Pul beads did not collapse at a basic pH, but they were swollen and deformed due to the ionisation of the COOH groups, which is in agreement with other authors [37], presenting a volume significantly higher than at the beginning of the experiment due to the repulsion among the charges at basic pH, as explained above. Therefore, there was a higher porosity of the beads, leading to a faster release. 

Cumulative release curves can provide valuable information on the release kinetics and mechanism. In Table 2, the goodness of fit of the different models was studied. The Higuchi model provided a relatively good fit for the experiments at pH = 7.

The Korsmeyer–Peppas model was used to explore the type of release mechanism. For *n* values <0.45, the drug transport release is considered quasi-Fickian diffusion, while for *n* values of ~0.45, the transport should be considered Fickian diffusion [53]. Cheng et al. [37], who used carboxymethyl chitosan/sodium alginate gels to encapsulate carvacrol, observed a Fickian release of the active principle at pH = 3, which accelerated and progressively became less Fickian as the pH of the buffer increased, resulting in a very fast release at basic pH. Wang et al. [46], who encapsulated curcumin in Alg beads with inclusions of CuO nanoparticles, observed a stronger dependence of the release on pH, with no release at pH = 2 over 2 h and a non-Fickian release at neutral pH due to the dissolution of the beads.

In Table 2, it can be observed that for the Alg-Ch beads, at pH = 2 and 7, the diffusion mechanism was Fickian. In these cases, it seems that the release of carvacrol was primarily diffusion from the core of the beads to the media through a well-formed Ch coating due to the strong COO−NH_3_^+^ interaction between Alg and Ch. This Ch layer caused the carvacrol molecules to diffuse though it before reaching the media. In the Alg beads, no external layer was present, and some carvacrol molecules were on the surface of the bead, so carvacrol on the surface released into the media quickly. For the Alg-Pul beads, the interaction between pullulan and alginate was weaker than the Alg-Ch interaction. Therefore, no Fickian diffusion was observed. However, the Alg-Pul beads provided a better behaviour at a high pH, since no disaggregation occurred because pullulan does not collapse at high pH values. The Pul layer slowed down the release of carvacrol in the Alg beads. 

At basic pH, the beads were more porous due to the swelling of the polymer, and they lost integrity. As a consequence, few models could fit the experimental behaviour.

Figure 8 shows the fitting of the Gallagher–Corrigan model to the release patterns at pH = 2.

Gallagher and Corrigan postulated, in their model [54], that a first release is attributed to the so-called burst effect. This effect is because some of the carvacrol molecules are encapsulated on the outer surface of the bead, and these molecules are rapidly released. The second stage corresponds to the slow diffusion of the carvacrol molecules from inside of the sphere. The Gallagher–Corrigan model properly fit the kinetics release while using the Alg beads, because during the encapsulation, alginate acts as a continuous phase of the carvacrol-loaded nanoemulsion, so some of the carvacrol molecules would be on the surface and, as consequence, would rapidly release into the media. 

The Ch and Pul slow down the burst effect. The Gallagher–Corrigan model is the best fitting model, but it should be mentioned that it is the model with more fitting parameters. When Ch is used as a coating material, the layer is better formed, as it is anchored more to the Alg core, with Fickian diffusion as the main release mechanism. 

## 4. Conclusions

In this work, carvacrol was nanoemulsified and further encapsulated to make it its incorporation in foods possible, as it has interesting antimicrobial and antioxidant properties that make it an alternative natural candidate.

The low-energy phase inversion method was successfully used to obtain carvacrol-loaded nanoemulsions using linoleic acid–potassium linoleate as the cosurfactant. The phasic behaviour of the carvacrol/coconut oil–linoleic acid–potassium linoleate)–/Tween^®^ 80–water system was studied for a neutralisation degree of 100%. Small-sized stable nanoemulsions could be obtained if a wide region without an excess of oil was crossed along the emulsification path, especially if this region extended towards a high % of water. Thus, as all of the oil was incorporated in this region, high energy was not required to disperse it when more water was added until a composition was attained in which the nanoemulsions were formed. Moreover, if the region without free oil extended nearer to the final composition of the nanoemulsions, less oil in excess existed in the nanoemulsion and small sizes were more easily obtained. The composition variables were studied, with the % (LA-KL) being a significant parameter regarding the nanoemulsions’ stability, as the HLB of the surfactant–cosurfactant mixture was relevant for the stability. Nanoemulsions were obtained with very small size, approximately 15–20 nm, and were stable for at least a 7-day period, which points to the small variation in the relative transmittance (experiments 4–7 in Table 1). The nanoemulsions were successfully loaded into gelled alginate beads. In order to delay the release of carvacrol from the alginate beads, two different coatings were added around these single Alg beads: chitosan and pullulan. The kinetics release profiles showed that the alginate beads presented a strong initial burst stage due to the high porosity of alginate and because the alginate and nanoemulsions were intimately mixed, so some % of the nanoemulsion was inevitably located on or near the surface of the beads and, therefore, poorly retained. When a second coating process was promoted by immersion of the Alg beads into solutions of chitosan or pullulan, the carvacrol release was more sustained over time, with the carvacrol diffusion mechanism in the media being Fickian diffusion when chitosan was used due to the chitosan layer around the alginate bead. The pullulan layer improved the delay of the release to a lower rate than that of chitosan, probably because the interaction between Alg and Pul was not as strong as Alg-Ch. However, the Alg-Ch beads collapsed at basic pH due to the dissolution of Ch that pulled the Alg of the beads because of the strong interaction between Alg and Ch, while the Alg-Pul beads provided a more sustained carvacrol release and protection of the beads. The Alg-Ch and Alg-Pul freeze-dried microspheres presented a more spherical shape and smoother surface than the ones obtained using only Alg. 

## Figures and Tables

**Figure 1 foods-12-01874-f001:**
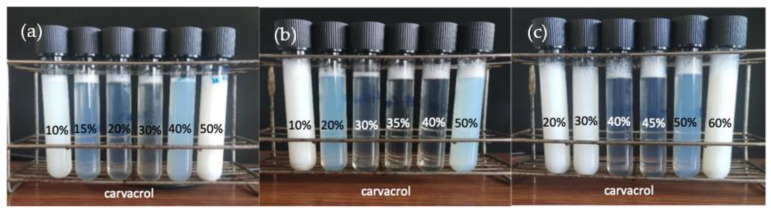
Preliminary experiments with 30% (LA-KL) and different % carvacrol/total oil: (**a**) 75% neutralisation degree; (**b**) 100% neutralisation degree; (**c**) 125% neutralisation degree. The final composition was 85% *w*/*w* water, 10% *w*/*w* surfactant mixture, and 5% *w*/*w* oil mixture.

**Figure 2 foods-12-01874-f002:**
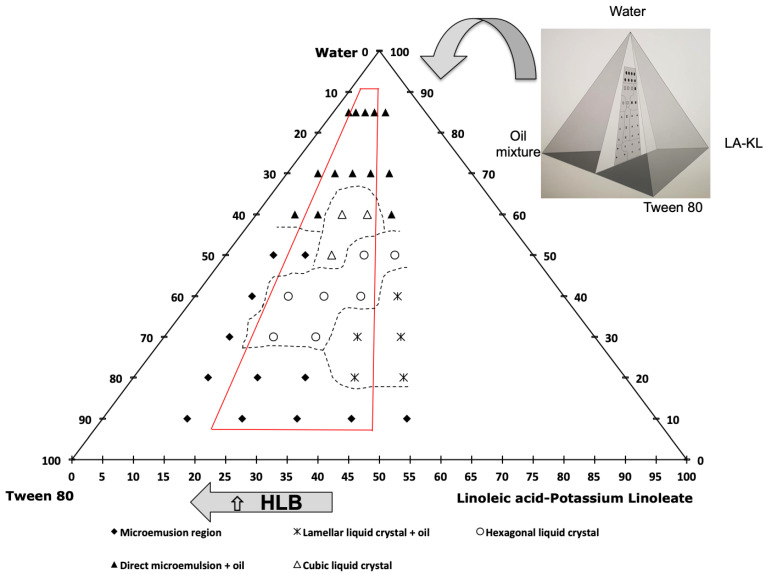
Phase diagram for carvacrol 25% *w*/*w* in the oil mixture, LA neutralisation degree of 100%, and (LA-KL) of 15%, 25%, 35%, 45%, and 55%. The oil mixture/(cosurfactant + surfactant) mass relationship was 0.5/1.

**Figure 3 foods-12-01874-f003:**
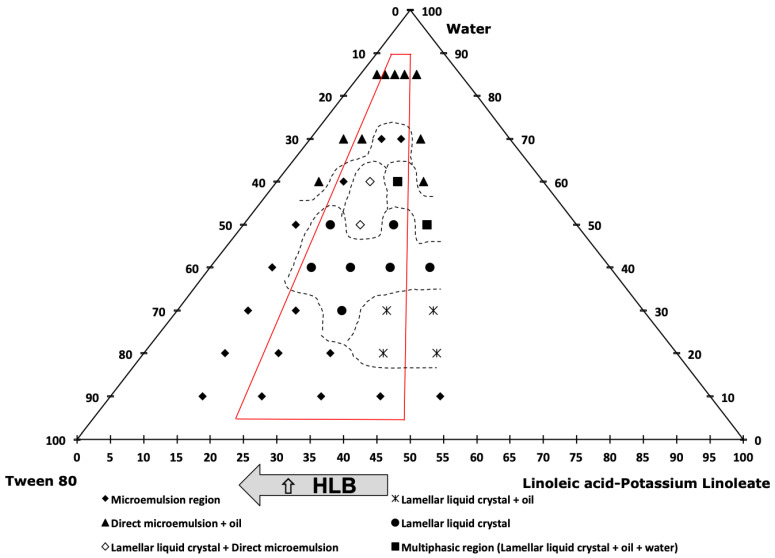
Phase diagram for carvacrol 35% *w*/*w* in the oil mixture, LA neutralisation degree of 100%, and (LA-KL) of 15%, 25%, 35%, 45%, and 55%. The oil mixture/(cosurfactant + surfactant) mass relationship was 0.5/1.

**Figure 4 foods-12-01874-f004:**
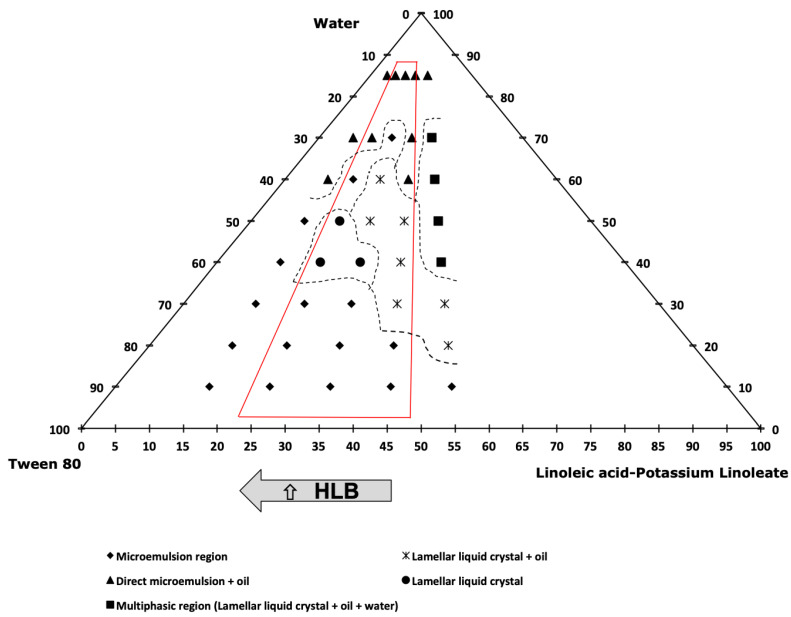
Phase diagram for carvacrol 45% *w*/*w* in the oil mixture, LA neutralisation degree of 100%, and (LA-KL) of 15%, 25%, 35%, 45%, and 55%. The oil mixture/(cosurfactant + surfactant) mass relationship was 0.5/1.

**Figure 5 foods-12-01874-f005:**
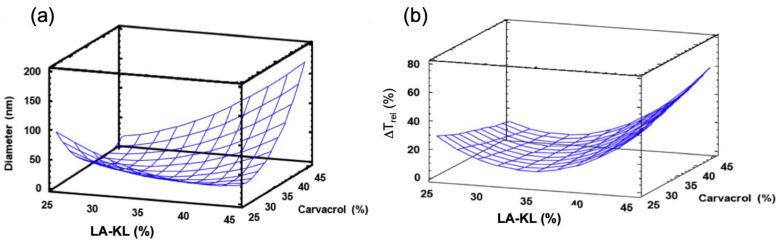
(**a**) Estimated response surface for the nanoemulsions’ mean droplet diameter; (**b**) estimated response surface for the nanoemulsions’ stability. The final compositions of all nanoemulsions were fixed at 5% *w*/*w* oil mixture, 10% *w*/*w* surfactant mixture, and 85% *w*/*w* water.

**Figure 6 foods-12-01874-f006:**
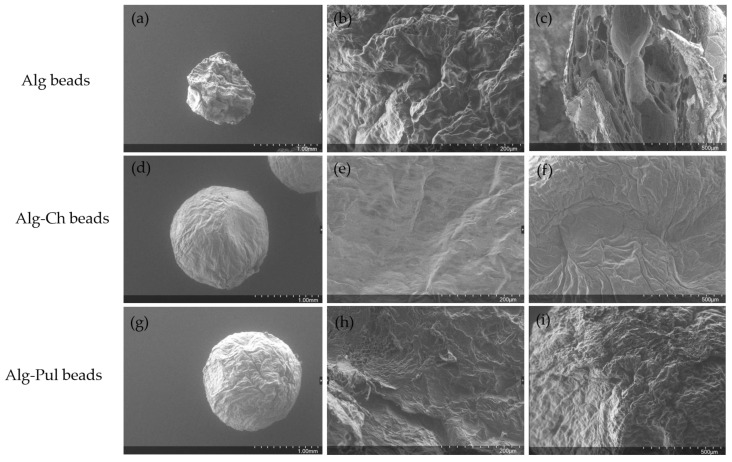
SEM micrographs at different magnifications: (**a**–**c**) Alg beads; (**d**–**f**) Alg-Ch beads; (**g**–**i**) Alg-Pul beads.

**Figure 7 foods-12-01874-f007:**
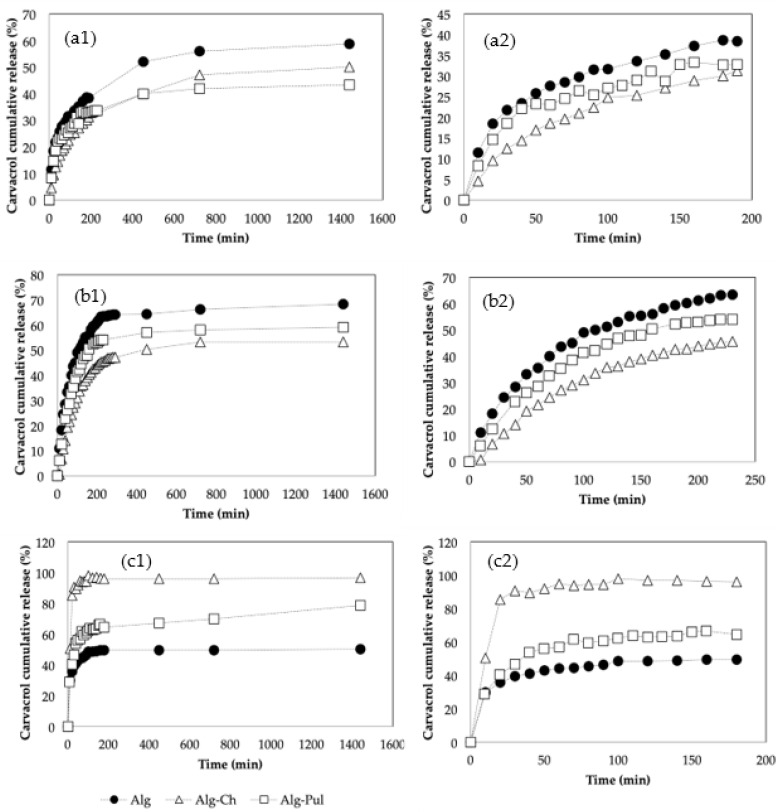
Cumulative carvacrol release of the Alg, Alg-Ch, and Alg-Pul beads at (**a**) pH = 2; (**b**) pH = 7; (**c**) pH = 12.

**Figure 8 foods-12-01874-f008:**
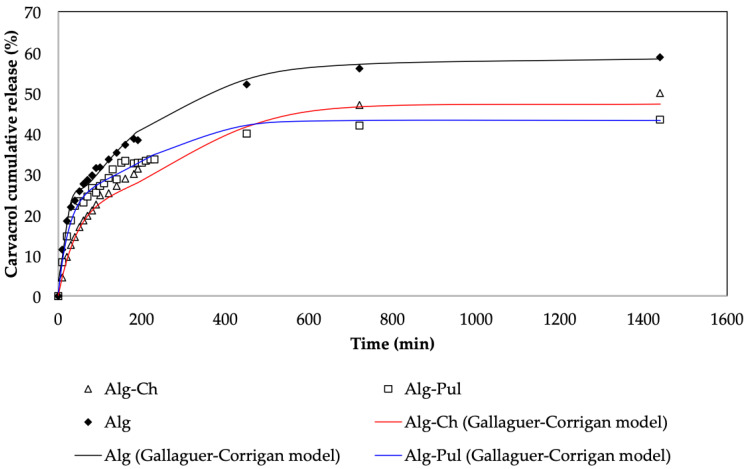
Gallagher–Corrigan model fitting for the Alg, Alg-Ch, and Alg-Pul beads at pH = 2.

**Table 1 foods-12-01874-t001:** Experiments carried out according to the experimental design. The composition variables were carvacrol (%) in the oil mixture and (LA-KL) (%) in the surfactant mixture. The response variables were the nanoemulsions diameter (D) and transmittance variation at 7 days (ΔT_rel_). The nanoemulsion polydispersity index (PDI) is also shown. The final oil mixture content (5% *w*/*w*), surfactant content (10% *w*/*w*), and water content (85% *w*/*w*) were fixed.

Experiment	Carvacrol (%)	LA-KL (%)	D (nm)	ΔT_rel_ (%) (7 Days)	PDI
1	25	25	74	35	0.64
2	25	35	57	11	0.42
3	25	45	32	30	0.31
4	35	25	21	9	0.26
5	35	35	17	4	0.15
6	35	35	16	6	0.17
7	35	35	18	4	0.16
8	35	45	13	39	0.40
9	45	25	32	15	0.39
10	45	35	21	20	0.38
11	45	45	209	47	0.72

**Table 2 foods-12-01874-t002:** Kinetics parameters of the carvacrol release from the Alg, Alg-Ch, and Alg-Pul beads at different pHs.

			Higuchi Model	Korsmeyer–Peppas Model		First-Order Model	Baker Model	Gallagher–Corrigan Model
Sample		M∞ (%)	R^2^	R^2^	*n*	R^2^	R^2^	R^2^
pH 2	Alg	58.79	0.882	0.938	0.317	0.915	0.964	0.986
	Alg-Ch	50.93	0.920	0.921	0.457	0.944	0.990	0.986
	Alg-Pul	43.36	0.868	0.956	0.283	0.845	0.915	0.999
pH 7	Alg	64.14	0.965	0.960	0.492	0.908	0.957	0.993
	Alg-Ch	47.20	0.971	0.960	0.458	0.915	0.970	0.989
	Alg-Pul	59.13	0.938	0.959	0.13	*	0.854	0.999
pH 12	Alg	94.39	*	0.960	0.170	*	*	0.988
	Alg-Ch	96.66	*	0.831	0.054	*	*	0.906
	Alg-Pul	91.83	*	0.892	0.162	*	*	0.890

* Indicates poor correlation.

## Data Availability

Data is contained within the article. Extra data will be provided on request.

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
