# Peer review of "Encapsulation of Carvacrol-Loaded Nanoemulsion Obtained Using Phase Inversion Composition Method in Alginate Beads and Polysaccharide-Coated Alginate Beads"

_foods, 2023, doi:10.3390/foods12091874_

Round 1
Reviewer 1 Report
This manuscript presents the preparation of nanoemulsions by low-energy emulsification phase inversion composition method using carvacrol, and studied the encapsulation and release of essential oil form alginate-based beads.
Several minor aspects were noticed.
1. The same verbal tense should be used in the Abstract.
2. Please check carefully the text for language, grammar, repetitions and typos. For example: “presented a moderately improve the release pattern of alginate beads.”; “distinctive characteristics property [29]. “This mixture was then hot into a test tube heater at 70 oC.”; “Nanoemulsions final composition was set to % w/w of oil mixture, 10% w/w of surfactant mixture” ; “to assure a property surface gelation.”;
3. Equation (1) –the notations could be simplified. The significance of “M” is not explained.
4. “For experiments 1-3, corresponding to phase diagram in Figure 2, the mean droplet size was smaller for (LA- KL) % w/w 3% and 45%.“ Probably we should read 35% instead of 3%?
5. “Experiment 10, with a 35% of (LA-KL), provided a small-sized nanoemulsion (21 nm) formation. In this case, a single direct microemulsion region was extended to water contents of 70% w/w.” Please re-phrase: for me it seems that for low water content, as well as for 70% microemulsion was obtained (Figure 4), while for 50% and 60% water, excess oil exists.
6. Table 1, legend: “Composition variables are carvacrol (%) ratio in the surfactant mixture”...- probably in the oil mixture. “Response variables are nanoemulsions diameter (D) and transmittance variation (ΔT rel). Nanoemulsion polidispersity index (PDI).” –please correct
7. “Smaller nanoemulsions are obtained when the phase without excess oil extends to higher percentages of water. It is probably related to the fact that, in these cases, less excess oil exists in the final formulation of emulsions, at 85%.” It is not clear for me: either there is a “phase without excess oil”, or “less excess oil”...
8. Clarification for the following phrase is needed: “According to phase diagram (Figure 4), it corresponds to the unique emulsification path with more extended region without oil excess to higher water %.” I believe the authors refer to the fact that 35% (LA-LK) was the only concentration where absence of oil was observed for high amount of water. However, this was a singular point, and not an “extended region”.
9. Why is it important to have sustained release of carvacrol? “For pH = 2, when pullulan and chitosan were used to coat the Alg beads, the release rate decreased, producing a slower, constant and more controllable release along time.” On the other hand, the amount of released active principle decreased... Which is more desirable: slower release or higher released amount? Please add a small discussion on practical implications of this release study.
10. Figure 7: please make the points more visible.
11. Table 2 is inserted in wrong position.
Author Response
We appreciate all the suggestions made by the reviewer and the time invested. Now we have considered the reviewer's comments and the responses and changes made are listed, each one below each suggestion:
Comments and Suggestions for Authors
This manuscript presents the preparation of nanoemulsions by low-energy emulsification phase inversion composition method using carvacrol, and studied the encapsulation and release of essential oil form alginate-based beads.
Several minor aspects were noticed.
- The same verbal tense should be used in the Abstract.
Thank you very much for your accurate revision. You are right that some verbal tenses are in the present and other in the past. All the tenses have been changed to the past one.
- Please check carefully the text for language, grammar, repetitions and typos. For example: “presented a moderately improve the release pattern of alginate beads.”; “distinctive characteristics property [29]. “This mixture was then hot into a test tube heater at 70 oC.”; “Nanoemulsions final composition was set to % w/w of oil mixture, 10% w/w of surfactant mixture” ; “to assure a property surface gelation.”;
We appreciate your considerations. Comments have been taken into account and the writing has been accurately revised.
- Equation (1) –the notations could be simplified. The significance of “M” is not explained.
The reviewer is right. We have arranged the equation. M is in fact Mco.
- “For experiments 1-3, corresponding to phase diagram in Figure 2, the mean droplet size was smaller for (LA- KL) % w/w 3% and 45%.“ Probably we should read 35% instead of 3%?
We are grateful for your comment. Of course, 3% is in fact 35%. We apologize for the mistake. We have changed it in the text.
- “Experiment 10, with a 35% of (LA-KL), provided a small-sized nanoemulsion (21 nm) formation. In this case, a single direct microemulsion region was extended to water contents of 70% w/w.” Please re-phrase: for me it seems that for low water content, as well as for 70% microemulsion was obtained (Figure 4), while for 50% and 60% water, excess oil exists.
Reviewer is correct that the sentence is confusing. We have rewritten it as follows: (lines 478-450)
“Experiment 10, with a 35% of (LA-KL), provided a small-sized nanoemulsion formation (21 nm). In this case, a single direct microemulsion region is found around water content of 70% w/w, while for the other (LA-KL) tested (15, 25, 45 and 55%) excess oil exists at 70% of water.”
- Table 1, legend: “Composition variables are carvacrol (%) ratio in the surfactant mixture”...- probably in the oil mixture. “Response variables are nanoemulsions diameter (D) and transmittance variation (ΔT rel). Nanoemulsion polidispersity index (PDI).” –please correct
We apologize again for the mistake. Of course, it is not “in the surfactant mixture”, but “in the oil mixture”. On the other hand, the response variables are just nanoemulsions diameter (D) and transmittance variation. Polydispersity index is shown in the table but it is not a variable response. The table legend has been rewritten for clarity as follows:
Table 1. Experiments carried out according to the experimental design. Composition variables are carvacrol (%) ratio in the oil mixture and (LA-KL) (%) ratio in the surfactant mixture. Response variables are nanoemulsions diameter (D) and transmittance variation (ΔTrel). Nanoemulsion polidispersity index (PDI) is also shown. Final oil mixture content (5% w/w), surfactant content (10% w/w) and water content (85% w/w) were fixed.
- “Smaller nanoemulsions are obtained when the phase without excess oil extends to higher percentages of water. It is probably related to the fact that, in these cases, less excess oil exists in the final formulation of emulsions, at 85%.” It is not clear for me: either there is a “phase without excess oil”, or “less excess oil”...
Reviewer is right that the sentence is unclear. At 85% of water, excess oil is always present. That is why a nanoemulsification is required. What we try to explain is that when a zone without excess oil is nearer to the final formulation (85% water), the amount of oil in excess at this final composition is less than when the single zone is far, and therefore it is easier to emulsify this oil in excess. We have rewritten the sentence as follows (466-471):
Smaller-sized nanoemulsions are obtained when the phase without excess oil extends to higher percentages of water. It is probably related to the fact that, in these cases, less excess oil exists in the final formulation of emulsions, at 85%. At 85% of water excess oil is always present. However, the amount of oil in excess is less when a zone with all the oil dissolved is near to the final composition than when it is far, therefore it is easier to emulsify.
- Clarification for the following phrase is needed: “According to phase diagram (Figure 4), it corresponds to the unique emulsification path with more extended region without oil excess to higher water %.” I believe the authors refer to the fact that 35% (LA-LK) was the only concentration where absence of oil was observed for high amount of water. However, this was a singular point, and not an “extended region”.
Thank you for your comment. The sentence has been changed because it is not an extended region, but the single region has high amount of water. The sentence has been rewritten (546-549):
According to Figure 4, for the emulsification path of 35% (LA-KL) all the oil remains dissolved at high amount of water, in contrast to the other paths tested for 45% of carvacrol. It also corresponds to the smaller droplet size obtained for 45% of carvacrol.
- Why is it important to have sustained release of carvacrol? “For pH = 2, when pullulan and chitosan were used to coat the Alg beads, the release rate decreased, producing a slower, constant and more controllable release along time.” On the other hand, the amount of released active principle decreased... Which is more desirable: slower release or higher released amount? Please add a small discussion on practical implications of this release study.
We appreciate the comment of the reviewer. According to his/her suggestion, we added the following discussion around lines 693-698:
Therefore, the burst effect was minimized by the layer of another polymer around Alg beads, protecting carvacrol for longer. As less carvacrol was released during the first 24 hours, the functional properties of beads conferred by carvacrol were presumably better maintained. The carvacrol retained into beads could act as a reservoir to provide sustained antimicrobial and antioxidant properties to the medium -for example a food- where beads could be added.
- Figure 7: please make the points more visible.
According to the suggestion of the reviewer, the points have been enlarged.
- Table 2 is inserted in wrong position.
Table 2 has been inserted in the proper place. It probably moved during uploading… We apologize.
Reviewer 2 Report
Authors prepared carvacrol nanoemulsion by phase inversion composition method and encapsulated the nanoemulsion in three beads. I recommend consideration of this work for inclusion in this issue, pending the clarification of the following.
(1) The list of “Alginate Beads, Alginate-Chitosan Beads and Alginate-Pullulan Beads” in the title is verbose.
(2) Potential application for the beads should be added in the abstract.
(3) Please add in the introduction what is the phase inversion composition method and its advantages.
(4) Line 66: Why use linoleic acid as a co-surfactant? what are its advantages please illustrate.
(5) Line 113: The degree of deacetylation of chitosan needs to be added.
(6) Line 211: What is the concentration of calcium chloride solution?
(7) Fig.1(a,b,c): It is necessary to explain why the concentration range and interval of carvacrol are different.
(8) Line 304: “as more carvacrol loaded into the nanoemulsions better antiox-idant and antimicrobial properties they will have”, here provide some references.
(9) Line 545: It is necessary to point out the interaction between Alginate and Pullulan, and provide more references to support “The layer is better formed when Ch is used, probably due to Ch is anchored by COO-NH3+ bonds to the Alg molecules forming a thicker layer than Alg-Pul”. Why the layer of Alg-Ch is thicker than that of Alg-Pul?
(10)Line 664: “stable at least during a 7 days period” experimental data is needed to support this conclusion.
(11)The spelling and grammar of the whole manuscript should be carefully improved.
(12)In the results and discussion, the authors should compare their results with existing studies and point out the merit of their gel beads.
Author Response
We appreciate the time invested by the reviewer and his/her valuable comments. We have accurately consider all of them and we have made the required changes. The comments are responsed point by point below:
Comments and Suggestions for Authors
Authors prepared carvacrol nanoemulsion by phase inversion composition method and encapsulated the nanoemulsion in three beads. I recommend consideration of this work for inclusion in this issue, pending the clarification of the following.
(1) The list of “Alginate Beads, Alginate-Chitosan Beads and Alginate-Pullulan Beads” in the title is verbose.
Thank you very much for your suggestion. In order to simplify the title, we have now changed it to:
Encapsulation of Carvacrol Loaded Nanoemulsion Obtained by Phase Inversion Composition Method in Alginate Beads and Polysaccharide-Coated Alginate Beads
(2) Potential application for the beads should be added in the abstract.
We appreciate the comment of the reviewer. The following sentence has been added to the abstract (lines 16-18):
Nanoemulsions with very small mean droplet diameter (16-20 nm) were obtained and successfully encapsulated to add carvacrol to foods as a natural antimicrobial and antioxidant agent.
(3) Please add in the introduction what is the phase inversion composition method and its advantages.
A brief explanation of the PIC method and its advantages has been added in the introduction as suggested by the reviewer (lines 59-73):
In order to form oil-in-water nanoemulsions (O/W) by PIC method, the components that will form the dispersed phase (oily components)+surfactants are first mixed and then the components that will form the continuous phase (aqueous components) are added progressively. For the PIC method it is required a change in the spontaneous curvature of the surfactant layer during the addition of the continuous phase. As a consequence, a phase inversion occurs. Several authors [16-18] reported that to obtain small-sized and stable nanoemulsions it is crucial to cross a region of direct or planar structure -liquid crystal or microemulsion- while adding the continuous phase, where no free oil appears. Thereby, the oil is already intimately incorporated in the structure. When more water is added to this structure, the spontaneous curvature increases, the structure reorganizes into droplets and the oil is entrapped in them without the requirement of high energy, resulting in small size diameters and narrow size distribution. Solè et al. [19] reported for their system smaller diameters using PIC method than using other high energy-consuming methods like sonication and high-speed homogenisation.
(4) Line 66: Why use linoleic acid as a co-surfactant? what are its advantages please illustrate.
An explanation of the advantages of partially neutralised linoleic acid as a co-surfactant is now included in the manuscript (lines 82-87):
The present study contributes to the PIC nanoemulsions formation using linoleic acid as a co-surfactant, a primary dietary omega-6 fatty acid. Linoleic acid is not synthesized by human body and needs to be incorporated into the organism in food. The linoleic acid role as a co-surfactant is crucial to form nanoemulsions by PIC method because the fatty acid increases its spontaneous curvature at the interface when it is neutralized by KOH forming the linoleate.
(5) Line 113: The degree of deacetylation of chitosan needs to be added.
Thank you for your observation. Deacetylation degree and molecular weight have now been included.
(6) Line 211: What is the concentration of calcium chloride solution?
Thank you for your appreciation. The concentration of CaCl2 solution has been added as follows (235-236):
“The method involved a two-stage procedure in which the beads were incubated into a 1.0% (w/v) CaCl2 solution…”
(7) Fig.1(a,b,c): It is necessary to explain why the concentration range and interval of carvacrol are different.
The use of different concentration range of carvacrol for the different neutralization degrees has been clarified as suggested (lines 313-318):
Nanoemulsions of small droplet size can be roughly identified since they are translucent or bluish. Therefore, the proper range of % carvacrol to be used can be found by eye, as when white aspect is observed the emulsions are coarse. It can be seen in Figure 1 that, as the neutralisation value increases, the carvacrol % w/w range that allows the formation of small-sized nanoemulsions, presenting a transparent-translucent appearance, moves towards higher values.
(8) Line 304: “as more carvacrol loaded into the nanoemulsions better antioxidant and antimicrobial properties they will have”, here provide some references.
We appreciate the suggestion of the reviewer. We agree that the addition of references will enrich the discussion of the manuscript. References included are:
[37] Cheng, M.; Cui, Y.; Guo, Y.; Zhao, P.; Wang, J.; Zhang, R.; Wang, X. Design of carboxymethyl chitosan-reinforced pH-responsive hydrogels for on-demant release of carvacrol and simulation release kinetics. Food Chemistry, 2023, 405, 134856. DOI: 10.1016/j.foodchem.2022.134856
[40] Vitali,A.; Stringaro, A.; Colone, M.; Muntiu, A.; Angiolella, L. Antifungal Carvacrol Loaded Chitosan Nanoparticles. Antibiotics, 2022, 11 (1), DOI: 10.3390/antibiotics11010011
[41] Loprestia, F.; Bottaa, L.; Scaffaroa. R.; Bilelloa, V,; Settannib, L.; Gaglio, R. Antibacterial biopolymeric foams: Structure–property relationship and carvacrol release kinetics. European Polymer Journal, 2019, 121, 109298. DOI: 10.1016/j.eurpolymj.2019.109298.
(9) Line 545: It is necessary to point out the interaction between Alginate and Pullulan, and provide more references to support “The layer is better formed when Ch is used, probably due to Ch is anchored by COO-NH3+ bonds to the Alg molecules forming a thicker layer than Alg-Pul”. Why the layer of Alg-Ch is thicker than that of Alg-Pul?
The referee is right, as we do not want to say that the layer is thicker, just that it is more anchored on the Alg beads due to the COO-NH3+ bonds. We have now added a reference and deleted the word “thicker” as follows (lines 612-615):
The layer is better formed when Ch is used, probably due to the fact that Ch is anchored by COO-NH3+ bonds to the Alg molecules [47-48] forming a more cohesive and uniform layer than Alg-Pul, which cannot have strong interactions.
(10)Line 664: “stable at least during a 7 days period” experimental data is needed to support this conclusion.
The sentence is supported by the relative transmittance changes showed in Table 1. Relative changes in transmittance is a measurement of stability of nanoemulsions. Reviewer was right pointing out that in Table 1 the period of time evaluated was not described. The correction can be found in Table 1 caption and in conclusions (Line 793-795):
“Nanoemulsions were obtained with very small size, around 15-20 nm, and stable at least during a 7 days period as points out the small variation in relative transmittance (experiments 4-7 in Table 1) .”
(11)The spelling and grammar of the whole manuscript should be carefully improved.
The authors have checked spelling and grammar of the whole manuscript.
(12)In the results and discussion, the authors should compare their results with existing studies and point out the merit of their gel beads.
Some discussion where more references have been added is now included in the manuscript. At lines 579-592 it has been added:
“As described in section 2.8, nanoemulsions were encapsulated in Alg beads coated by different materials. The beads were formed using the same Encapsulator (Buchi B-390) and the same operating variables than Atencio et al. [29]. The mean diameter was ~1600 m (p < 0.05). No significant differences between mean values were observed for uncoated and coated beads (p < 0.05). It is well reported [29,37,46, 47] that single Alg gel beads have a high permeability and porosity and poor mechanical properties that keep down their protection and control release ability. In order to improve the Alg hydrogels, Wang et al. [46] proposed the inclusion of carboxymethyl-chitosan-copper oxide nanoparticles in the Alg matrix. Cheng et al. [37] reported the use of a mixture of carboxymethyl chitosan and sodium alginate, crosslinked by citric acid and subsequent cooling to -80ºC. Both studies obtained an improvement of protection properties and a slower release, that were pH dependent. In fact, the release accelerated at basic pH due to swelling and disruption of the matrix. In the present work, the Alg beads were reinforced by a simpler method, based on [29]..”
And at lines 734-754 it can now be read:
Cheng et al. [37], who used carboxymethyl chitosan/sodium alginate gels to encapsulate carvacrol, observed a Fickian release of the active principle at pH = 3, that accelerated and progressively became less Fickian as the pH of the buffer was increased finding a very fast release at basic pH. Wang et al. [46], who encapsulated curcumin in Alg beads with inclusions of CuO nanoparticles, observed a stronger dependence of release with pH, with no release at pH = 2 during 2 hours and a non-Fickian release at neutral pH due to the dissolution of beads.
In Table 2, it can be observed that for Alg-Ch beads, at pH = 2 and 7 the diffusion mechanism is Fickian. In these cases, it seems that the release of carvacrol is primarily diffusion from the core of the beads to the media through a well-formed Ch coating due to the strong interaction between Alg and Ch. This Ch layer causes that carvacrol molecules must diffuse though it before reaching the media. In Alg beads no layer is present and some carvacrol molecules are on the surface of the bead, so carvacrol on the surface releases to the media quickly. For Alg-Pul beads, the interaction between pullulan and alginate is not as well built as Alg-Ch. Therefore, no Fickian diffusion was observed. However, the Alg-Pul beads provided a better behaviour at high pH, since no disaggregation occurred because pullulan does not collapse at high pH values. The Pul layer slowed down the carvacrol release of Alg beads.
At basic pH the beads are opened due to the polymer swelling and they lose integrity. As a consequence, few models can fit the experimental behavior.
Reviewer 3 Report
Highlight in the introduction the importance of carvacrol, why encapsulate it?
The experimental design used in the methodology is not described, please explain it.
The author mentions that samples smaller than 20 nm were obtained, how does he verify this?
Improve figure 5.
In point 3.3, the author mentions the porosity of the alginate, can you compare the porosity with the other materials and how is it related to the release of carvacrol?
In figure 7, I recommend that the author not zoom in and put it inside the figure, rather make another figure up to 200 min and put them aside, the release will be better seen.
the COO— of alginate was transformed into –COOH, and the ionization degree was reduced, any evidence of this?
Figure 8, why only at pH 2?,
You didn't do particle size, you should.
As well as pore size.
Author Response
We appreciate the time invested by the reviewer and his/her valuable comments. We have accurately consider all of them and we have made the required changes. The comments are responsed point by point below:
1.- Highlight in the introduction the importance of carvacrol, why encapsulate it?
Thank you for your comment. The importance of carvacrol has been clarified in lines (39-50):
“Among the best-known oils are carvacrol, thymol, eugenol, linalool, cinnamaldehyde, D-limonene, and others. Each of these is mostly found in a different plant and has different characteristics and properties [2-3]. Carvacrol has been proved as one of the EO with the strongest antimicrobial and antioxidant properties. Since this compound has a very low solubility in water, a way to add it in aqueous media is in the form of an oil-in-water emulsion, O/W. Nanoemulsions can be defined as emulsions with a droplet size up to 200 nm and low polydispersity, presenting more stability than conventional macroemulsions. These are used to entrap the EO, finely disperse them in the medium, increase their stability in the environment and minimise the change in the sensory properties of the products, especially for food preservation and functionalisation [4]. The encapsulation of these nanoemulsions into gel beads would immobilise them and offer a release control to the EO.”
2.- The experimental design used in the methodology is not described, please explain it.
The reviewer is right. The experimental design has been described in lines 426-436:
“An experimental design was used in order to obtain information about which composition variable had a significant influence in the obtaining of stable, small diameter nanoemulsions, and in order to obtain an optimal formulation regardind droplet size and stability. As the phase diagrams showed, multiple variables could affect the resulting nanoemulsion. Experimental design provided an easiy way to find out the influence of each variables and their possible interaction, as the optimum of each variable can move depending on the level of the other variable. A factorial 3k + central point design was carried out, the number 3 indicating that three levels were studied for each variable, and where k = 2 was the number of input variables studied, i.e. the carvacrol % (from 25% to 45%) in the oil mixture and (LA-KL)% (from 25% to 45%) in the surfactant mixture. The central point was replicated 3 times. “
3.- The author mentions that samples smaller than 20 nm were obtained, how does he verify this?
Thank you for your comment. The droplet size of the nanoemulsions was measured using Dinamic Light Scattering. For each simple the result shown in table 1 is the mean value of three independent measurements. The methodology is explained in section 2.5:
Dynamic light scattering measurements were performed by the Nanostructured Liquid Characterization Unit. 3D Dynamic light scattering (3DDLS) spectrometer (LS Instruments, Switzerland) equipped with a He-Ne laser (λ=632.8 nm) was used, allowing the temperature control (25oC) at a scattering angle of 90°. The light intensity correlation function was analysed through on the multimodal method, whereas the z-average diameter was obtained by cumulant analysis. The reported droplet size of nanoemulsions was the mean of at least three independent measurements.
4.- Improve figure 5.
The figure has been improved, and the Y axis of figure 5b has been set right.
5.- In point 3.3, the author mentions the porosity of the alginate, can you compare the porosity with the other materials and how is it related to the release of carvacrol?
We appreciate the comment of the reviewer. In fact, we have not measured the porosity. However, it has been reported a high porosity and/or a high permeability of Alg gels that can be reduced by coating it with other polysaccharide of by other methods now described in the manuscript. We have added references as follows (lines 582-592):
“The beads were formed using the same Encapsulator (Buchi B-390) and the same operating variables than Atencio et al. [29]. The mean diameter was ~1600 mm (p < 0.05), measured by microscopy. No significant differences between mean values were observed for uncoated and coated beads (p < 0.05). It is well reported [29,37,46, 47] that single Alg gel beads have a high permeability and porosity and poor mechanical properties that keep down their protection and control release ability. In order to improve the Alg hydrogels, Wang et al. [46] proposed the inclusion of carboxymethyl-chitosan-copper oxide nanoparticles in the Alg matrix. Cheng et al. [37] reported the use of a mixture of carboxymethyl chitosan and sodium alginate, crosslinked by citric acid and subsequent cooling to -80ºC. Both studies obtained an improvement of protection properties and a slower release, that were pH dependent. In fact, the release accelerated at basic pH due to swelling and disruption of the matrix. In the present work, the Alg beads were reinforced by a simpler method, based on [29].”
6.- In figure 7, I recommend that the author not zoom in and put it inside the figure, rather make another figure up to 200 min and put them aside, the release will be better seen.
Thank you for your recommendation. We arranged the Figure 7 as suggested.
7.- the COO— of alginate was transformed into –COOH, and the ionization degree was reduced, any evidence of this?
Thank you for your consideration. The discussion has been enriqued. As the reviewer pointed out, the reduction of ionization degree has been evidenced by other authors as cited in lines 676-384 by three different authors.
“Nazli & Açikel [48] obtained that the resveratrol released from alginate beads was positively correlated with pH, being the release rate higher as pH increases. This was attributed to the fact that, under acidic conditions, the COO— of alginate was transformed into –COOH, and the ionization degree was reduced [48], which led to the contraction of alginate molecular chain, so that the shell of the hydrogels was dense and retarded the release more efficiently. Whereas, the degree of dissociation of Alg –COOH groups increased with increasing pH, which leaded to stronger repulsion between single –COO— groups. This behaviour promoted the swelling of alginate beads and an increase of porosity. This pattern was confirmed in other studies [37,50-51]”
And lines 706-713:
“Alg and Alg-Pul beads did not collapse at basic pH, but they were swollen and deformed due to the ionization of the COOH groups, in agreement with other authors [37], presenting a volume significantly higher than at the beginning of the experiment due to the repulsion between charges at basic pH, as explained above.”
8.- Figure 8, why only at pH 2?,
In figure 8 only one pH was chosen in order to illustrate the Gallaguer-Corrigan fitting curve. The authors considered that more pH patterns would be redundant for this purpose.
9.- You didn't do particle size, you should. As well as pore size.
The reviewer is right. Particle size was not included in the manuscript. The beads were formed using the same experimental method than Atencio et al. [29]. The mean size of the beads has been included in lines 581-583 supported by the appropriate reference.
“As described in section 2.8, nanoemulsions were encapsulated in Alg beads coated by different materials. The beads were formed using the same Encapsulator (Buchi B-390) and the same operating variables than Atencio et al. [29]. The mean diameter was ~1600 mm (p < 0.05). No significant differences between mean values were observed for uncoated and coated beads (p < 0.05). “
The pore size has not been measured. However, the high porosity and permeability of Alg beads has been reported by several authors, who improved them by several techniques, including the combination with other polysaccharides like Chitosan. Now we think that have responded to this question when we responded the question 5 of the same reviewer.
Reviewer 4 Report
In this manuscript, the authors prepared nanoemulsions successfully encapsulated into alginate beads by external gelation.
There are a few major comments as follows:
1. In the last section, it is recommended to include the rationale of the research.
2. In the material section, include the molecule weight of each component.
3. in the 2.4 section, it is recommended to include the detailed procedure of formulation.
4. Have authors carried out the drug release study in duplicate?
Author Response
We appreciate the time invested by the reviewer and his/her valuable comments. We have accurately consider all of them and we have made the required changes. The comments are responsed point by point below:
In this manuscript, the authors prepared nanoemulsions successfully encapsulated into alginate beads by external gelation.
There are a few major comments as follows:
- In the last section, it is recommended to include the rationale of the research.
Thank you for your suggestion. Now we have included the following sentence to the last section (lines 776-778):
In this work, carvacrol is nanoemulsified and further encapsulated to make it possible its incorporation in foods, as it has interesting antimicrobial and antioxidant properties that make it an alternative natural candidate.
- In the material section, include the molecule weight of each component.
The comment of the reviewer has been attended and the molecular weight of each component has been included as follows (Line 131-138):
“chitosan (419419) > 75% deacetylated degree and high MW 3.1-3.75x105 Da, calcium chloride (C1016) anhydrous, granular, ≤ 7 mm with purity ≥ 93%, Sudan IV (198102) and Brilliant Blue G (27815) were purchased from Sigma Aldrich. Technical grade sodium alginate (Alg) with a ratio β-D-mannuronic acid:α-L- guluronic acid = 58.9:41.1, measured using nuclear magnetic resonance (DMX-500, 500 MHz, Bruker, Billerica, MA, USA), and Mn ≈ 668,000, Mw ≈ 1,750,000, obtained using size exclusion chromatography (see below for methods details) was purchased from Panreac. Pullulan low MW 4.2x105 Da from ITW reagents was used.”
- in the 2.4 section, it is recommended to include the detailed procedure of formulation.
We appreciate the comment of the reviewer. Now, the procedure of formulation in section 2.4 has been detailed including the paragraph (lines 161-167):
For the study of nanoemulsion formation, 10 g of each sample was prepared. The final water concentration of the emulsions was fixed at 85% w/w, the co-surfactant-surfactant mixture (linoleic acid-potassium linoleate)-Tween 80® ((LA-KL)-T80) at 10% w/w and the oil mixture, carvacrol-coconut oil, 5% w/w. First, 1 g of the mixture of surfactants+co-surfactant (Tween 80 ® and linoleic acid) at the desired composition was weighed, in the range 25%-45% (LA-KL). Next, 0.5 g of the oil mixture, composed by different carvacrol% in a carrier of coconut oil (25%-45%), was added.
- Have authors carried out the drug release study in duplicate?
The reviewer is raight, as we have not clarified this point. Release experiments for pH=7 were carried out by duplicate as well as some experiments for pH=2 and pH=12. The lines 621-625 have been included in the manuscript in order to answer the reviewer’s question:
“The drug release study was carried out in duplicate for all the experiments at pH = 7 and for one experiment at pH = 2 and one at pH = 12 in order to evaluate the reproducibility of the patterns. The cumulative release results of duplicates were similar, indicating that results are reproducible.“
Round 2
Reviewer 3 Report
The authors have addressed the comments, it can be published.
Reviewer 4 Report
The revised manuscript can be considered for further journal process.